# Are Stock Markets among BRICS Members Integrated? A Regime Shift-Based Co-Integration Analysis

**Ayesha Siddiqui** [1,2] **, Mohd Shamim** [1]**, Mohammad Asif** [3] **and Mamdouh Abdulaziz Saleh Al-Faryan** [4,*]

1 Department of Commerce, Aligarh Muslim University, Aligarh 202001, India;
   ayeshasiddiqui.91@gmail.com (A.S.); shamim1234@gmail.com (M.S.)
2 Universal Business School, Karjat, Mumbai 410201, India
3 Department of Economics, Aligarh Muslim University, Aligarh 202001, India; m.asif.ec@amu.ac.in
4 Department of Accounting and Financial Management, University of Portsmouth, Portsmouth PO1 3DE, UK
* Correspondence: al-faryan@hotmail.com

**Abstract:** Long-run relationships and structural breaks have often been confused so that many investigators ignore the structural breaks in long-run stock price relationships. In this paper, we investigate the long-run relationships among stock prices in BRICS countries in a bivariate framework. We used a non-linear threshold cointegration test, which endogenously incorporates possible regime shift behaviors into the long-run relationships from 2004 to 2018. The Johansen cointegration test, the Gregory and Hansen cointegration test, and the Hatemi-J regime shift cointegration test, which allow for single and double structural breaks, were used. The principal finding of this paper confirms the presence of cointegration among the BRICS stock markets with two endogenous structural breaks. The study confirms that ignoring the presence of structural breaks in long-run series data can produce ambiguous results. It also confirms the absence of cointegration among these stock markets (Brazil and China, India and China, and China and South Africa) after two endogenous structural breaks. These empirical findings support conjecture on more than just the changes in the relationships between the BRICS stock markets. The disintegrated markets suggest the absence of arbitrage activity and vice versa. Thus, disintegrated markets mean that investors can obtain long-term gains through international portfolio diversification. While the benefit of the diversification is very limited in the long run, it is unlikely to be eliminated in practice. Hence, there is a possibility of obtaining an unusual profit in such a market, and consequently the assumptions of market efficiency could also be violated.

**Keywords:** endogenous structural break; BRICS stock markets; diversification; Gregory–Hansen; Hatemi-J; Johansen cointegration



## 1. Introduction

The expansion of economic globalization has led to interconnection, trade, and capital flow among the five major developing economies; those of Brazil, Russia, India, China, and South Africa, commonly known as BRICS (Sui and Sun 2016; Coulibaly et al. 2018; Armijo 2007). The BRICS members remain the beneficiaries of vital investment globally, and they are leading trading partners with the USA, Japan, and Germany, and thus their stock markets are closely connected with these major economies (Garcia and Bond 2018; Yu 2017). They are expected to exhibit exceptionally high economic growth in the coming 50 years as a result of becoming more economically sound than the G6 countries (Wilson and Purushothaman 2003a, 2003b; Burrascano 2018).

With the easing of capital controls in the BRICS member states, investors' interest in international diversification has amplified. International diversification of portfolio assets improves the risk–reward ratio (Nashier 2015; Kajtazi and Moro 2017; Saritas and Aygoren 2005). Nevertheless, when equity markets are cointegrated, the advantages of international

diversification remain limited. Common factors limit the amount of independent variation in stock markets (Wong et al. 2005; Nashier 2015; Cheng et al. 2007). Integration among stock markets implies fewer assets being available to investors for portfolio diversification.

Moreover, cointegration also suggests inefficiency in markets (Richards 1995; Wong et al. 2005). With globalization, investors have developed active interests in overseas capital marketplaces. International financial market investments have seen a huge increase in recent years (Santiso 2008; Dunning et al. 2007; Dunning 2009). Financial markets become integrated under market liberalization, and this integration process suggests bigger co-movements between financial markets that can have undesirable effects on the benefits of international diversification. Nonetheless, the financial crisis in the United States of America led investors to search for other developing markets, e.g., the emerging markets of BRICS. These markets can help by providing surplus opportunities to upsurge benefits from international diversification.

The efforts to get these economies in line with Western developed nations has led investors to study their investment offerings (Fadhlaoui et al. 2009; Nashier 2015). The great level of integration amongst international markets calls for conducting comprehensive studies on the integration between developed and emerging markets to highlight the significant potential of emerging markets for international portfolio diversification (Bekaert and Harvey 2003; London and Hart 2004). We thought that it would also be interesting to examine how long-term relationships between international financial markets have emerged. This paper investigates whether there is a long-term relationship in the regime-switching behavior between the stock markets of the emerging economies of BRICS (Brazil, Russia, India, China, and South Africa).

This study examines whether integration and long-term causal relationships exist between BRICS stock markets. This paper contributes and differs from the extant literature in five ways: (a) We examine whether the break in stock markets occurred simultaneously or whether there were variations. (b) We provide evidence of whether regime-switching behavior is prevalent in these markets. (c) We examine the effects that regime-switching behavior has on the long-term relationships of stock markets, which the linear conventional cointegration model cannot capture. (d) We provide evidence that financial crises aggravated the effects between the BRICS stock markets and endogenous regime shifts, causing nonlinearity in the relationships, considering single and double structural breaks. Understanding and estimating the endogenous regime-switching behavior effects have significant consequences on asset allocation, portfolio diversification, and stock market return predictability. (e) We also employed a single linear model that incorporates a dummy to simultaneously examine the intercept, trend shift, and regime shift in order to check the impacts of single and double structural changes on BRICS stock indices from 2004 to 2018.

Accordingly, we studied the presence of a structural break in the long-term relationships of BRICS stock markets. We employed Gregory and Hansen's (1996a, 1996b) endogenous single structural break model to identify the impact of the crisis on the relationships between stock markets. We also employed Hatemi-J's (2008) double structural break model to identify the presence of integration from 2004 to 2018. Our modest model incorporates a dummy. Using it, we simultaneously examined the intercept, trend shift, and regime shift in a single linear equation, assessing the impacts of structural changes in BRICS stock indices from 2004 to 2018.

The paper has five sections. The literature is discussed in Section 2. Section 3 deals with the data and methodology, while Section 4 contains the data analysis and interpretation. The conclusions and implications of the study are discussed in Sections 5 and 6, respectively.

## 2. Literature Review

The literature review has been divided into five sections with a view to providing detailed insights on various aspects of the concept of regime-switching behavior and its impact on stock markets.

## 2.1. The Concept of Structural Break

In finance, the testing, identification, and analysis of structural breaks in a time series analysis have grave significance for econometric modelling (Hansen 2001). Identifying structural change is vital for any model to capture the correct estimated relationships between the variables (Kalsie and Arora 2019). The effect of any break in the market is seen when any crisis occurs during the timeline. Stock markets worldwide are interconnected due to globalization and financial integration. For example, the global financial crisis (hereafter, GFC) that began in the USA on 2nd April 2007 affected almost all the world's financial markets (Chen et al. 2018). The spread of hazards and the complex character of external and internal events in a local stock market necessitate a detailed examination of stock correlation networks and structural dynamics. In the financial network literature, correlation-based networks are commonly employed to measure the impact of various crises' occurrences (Kashi et al. 2019).

## 2.2. Long-Run Relationship of Stock Markets and Macroeconomic Variables

The long-run relationship between crude oil prices and the international stock market, while allowing for short-run macroeconomic influence on stock prices, has been studied by Miller and Ratti (2009). However, this work contrasts with other studies focused on the short-term impact of oil prices on stock market returns (Nandha and Faff 2008; O'Neill 2002). The issue of the long-run relationship between markets has been extensively examined considering different aspects (see, e.g., Sehgal et al. 2015; Ghosh and Kanjilal 2016; Nambiappan et al. 2018). Salami and Haron (2018) studied the long-run relationship by taking into account the speed of adjustment of short-run deviations in the price efficiency and found that once the long-run relationship between the two markets is established, market deviation in the short-run becomes temporary.

There is substantial literature examining the behavior of stock prices (e.g., Balvers et al. 2000; Buguk and Brorsen 2003), and some researchers have looked at the effect of the macroeconomic variable on the stock market while taking a structural break into account (Kalsie and Arora 2019; Yurdakul and Akçoraoğlu 2003; Chifurira et al. 2016). Prior studies have investigated the impact of macroeconomic variables in the presence or absence of a structural break. In the same vein, Sui and Sun (2016) examined the dynamic relationships between local stock returns, foreign exchange rates, interest differentials, and USA S&P 500 returns while looking at the spillover effect between the USA and BRICS stock and foreign exchange markets. Meanwhile, Narayan et al. (2010) used the common structural break test of Bai and Perron (1998) to test the effect of a common structural break on American, British, and Japanese stock prices.

## 2.3. Co-Movement or Integration among the Markets

Movement in the stock market is dependent on the integration of the market (Salami and Haron 2018). In the finance literature, the issue of financial market integration is of crucial importance in both theoretical and practical terms. Investors in the international market need information about the integration of the financial market to determine and measure the amount of risk to mitigate the risk in their portfolios. Thus, one of the basic principles of portfolio diversification is to construct a portfolio with uncorrelated returns between financial markets (Liu et al. 2013). Integration or co-movement between markets is termed strong information flow, which is related to market efficiency. The efficient market hypothesis states that the stock market fully reflects all the information available in the form of an increase or decrease in the stock price and that it follows a random walk process (Billio and Pelizzon 2003; Cevik et al. 2017).

Evidence favoring stock market integration suggests that there is a lack of efficiency in the market since the presence of integration between markets implies co-movement between markets (see Yarovaya and Lau 2016; Prakash et al. 2017; Hamulczuk et al. 2019). However, dynamic relations within a stock market can help policymakers to formulate policies to safeguard the stock market against the contagion effect of a structural break in

the economy. Studies on stock markets have examined the presence of integration between stock markets using different methods (Kalsie and Arora 2019; Sui and Sun 2016; Narayan et al. 2013). Long memory in the stock market in general, and the asset price or stock price in particular, implies that price changes are heavily dependent on price changes in the distant past. Thus, it is possible for future price changes to be predicted from past price changes. If investors have prior knowledge regarding the persistent stock price changes, they can reap huge profits by buying or selling stocks when prices are expected to rise or fall from the mere observation of the persistence of price behaviors (Ngene et al. 2017).

### 2.4. Co-Movement among the Stock Markets during Crisis Periods

There is a different strand of literature that studies co-movements between developed markets; each one focuses on a single aspect. A study by Ratanapakorn and Sharma (2002) examines the long- and short-run relationship between five regional stock indices using the Johansen cointegration test and a VAR model in the pre- and within-Asian crisis periods. In addition, Nieh and Lee (2001) and Chen et al. (2002) both used the conventional cointegration technique to identify the effect of the Asian financial crisis on several East Asian economies and to study the interdependence of stock markets in Latin America, respectively. Sheng and Tu (2000), Pierdzioch and Kizys (2012), Click and Plummer (2004), and Lee and Zeng (2011) studied integration during the Asian financial crisis among the NAFTA countries and the ASEAN stock market using the traditional cointegration technique. Other studies that incorporate the cointegration test to study integration and crisis impacts between countries include Yang et al. (2006) and Koop and Korobilis (2016).

### 2.5. Dynamic Relationship between BRIC/BRICS Stock Markets

The studies conducted by Chittedi (2009), Bhar and Nikolova (2009), and Aloui et al. (2011) explore the dynamic relationships and co-movements between the BRIC/BRICS countries. Tripathi and Kumar (2014) used the panel cointegration technique to study this topic. In addition, Lian and Brown (2010) studied the co-movement between USA and BRIC stock returns, and Yarovaya and Lau (2016) studied the co-movement between the BRICS and MIST emerging markets. In comparison, Prakash et al. (2017) used monthly data to study the degree of financial integration among the BRICS countries. Most of the studies discussed above show no cointegration relationship between the markets. In addition, Sui and Sun (2016) studied the dynamic relationships between local stock returns, foreign exchange rates, interest differentials, and USA S&P 500 returns. Meanwhile, Chkili and Nguyen (2014) showed that the unilateral impact of the stock market on the foreign exchange market was significant for all BRICS members during periods of the high volatility, except South Africa, using Markov switching VAR models. However, thus far few studies have incorporated single and double structural break non-linear cointegration models to study the concept of integration and to look at the impact of regime-switching behavior on the BRICS stock markets.

### 2.6. Level of Integration in the Presence of a Structural Break

This strand of literature includes studies that have studied the level of integration between stock markets and their impact on structural change. Bodhanwala et al. (2020) explored the effect of structural change on the flow of information between spot and futures markets and selected macroeconomic variables. Singh and Sharma (2018) investigated the cointegration and causal relationships between gold, crude oil, US dollars and Indian stock market during the global financial crisis of 2008 using the Johansen cointegration test, VECM, VAR, the Granger causality test, and the variance decomposition test. Shahbaz et al. (2016) used the macroeconomic determinants of stock markets in Pakistan to study the integrating properties between the variables using the ARDL bound test. The studies highlighted above show that all the variables are cointegrated. Narayan et al. (2013) used the common structural break test propounded by Bai and Perron (1998) to test the effect of a common structural break on American, British, and Japanese stock prices and

investigated whether structural break has slowed down the growth of stock markets. The average annual growth suggests that the structural break has slowed down the growth rate of the American, British, and Japanese stock markets. Many empirical studies in the literature have shown the impact of macroeconomic variables on the stock market when incorporating a structural break (Wei et al. 2019; Ji et al. 2020; Salami and Haron 2018; Miller and Ratti 2009).

The conventional cointegration test has been criticized because it has low power to reject the null hypothesis in the presence of regime-switching behavior in the series according to studies that focus on a structural break while studying the long-run movement in stock markets. For instance, a study by Yavuz (2014) highlights the cointegration of variables with a single structural break. The variables used in the study were $CO_2$, energy consumption, and economic growth in Turkey. Ghosh and Kanjilal (2014) used a nonlinear cointegration test to study the co-movement of international crude oil prices and the Indian stock market. The period of the study was from 2003 to 2011. The study does not show any long-run integration among the variables during the study period. To better understand the results, the entire period of the study was further bifurcated into three sub-periods, where cointegration was present only in the third phase. Luo and Huang (2018) explored long memory and structural breaks in stock index volatility series.

Another study by Kanjilal and Ghosh (2013, 2014) investigated the relationship between the gold import demand, gold price, and Indian GDP. The study results show that gold import demand is moderately inelastic to unitary elastic with respect to gold price in the long run. Gregory and Hansen (1996a, 1996b) and Hatemi-J (2008) tests have been used to study the relationship between Turkey's stock market and bond market. The government bond index and stock market indices are not cointegrated (Evrim-Mandaci et al. 2011). Another study looked at the effect of a structural break on the volatility spillover between the five major Latin American markets (Güloğlu et al. 2016). The study results show that volatility spillover effects between markets are not strong. In addition, causality in mean indicates one-way causality from BOVESPA to all markets, whereas causality in variance indicates one-way causality only from BOVESPA to IPSA. The long-run relationship of the stock market has been studied considering macroeconomic variables (Lin 2012). Okere et al. (2021) studied the symmetric and asymmetric effects of crude oil price and exchange rate on stock market performance using multiple structural breaks and NARDL analysis.

Ngene et al. (2017) studied the long-run dependency between the exchange rate and the inflation rate using the weekly return in seven African stock markets. Their results show that the variable long-term memory declined monotonically in magnitude. Their results indicate that long-memory components vary in their returns without a break. Fowowe (2014) examined the relationship between the stock prices and exchange rates in South Africa and Nigeria while incorporating a structural break in the long-run relationship. Using Gregory and Hansen's single structural break model, Mohti et al. (2019) investigated the level of regional and global market integration among the emerging and frontier Asian countries. All the emerging markets showed integration at both regional and global levels, whereas in the case of the frontier markets this is true in Pakistan only.

In addition, studies on integration with an endogenous structural break on the BRICS stock markets have been conducted. In a study on the BRICS stock markets and crude oil, Wang et al. (2020) studied the average causal relationship with an extreme Granger causality analysis model. The results of their study show that the effect of oil price changes on the stock markets is stronger under extreme circumstances than under normal circumstances.

### 2.7. Summing Up the Findings from the Literature

As can be seen from the literature above, much work has been conducted to document co-movement between international stock markets. However, little work has been carried out examining the cointegration of stock markets while incorporating a structural break in general and looking at macroeconomic variables in particular. There have been limited studies investigating the long-run relationships between the BRICS stock markets that

incorporate regime-switching behavior. With the abovementioned limitations in mind, studying regime-switching behavior in the BRICS stock markets is imperative. Our study has been conceptualized and designed in order to fill the gaps that exist in the extant literature.

### 3. Methodology

*3.1. Data Description and Econometric Methodology*

The data used in this study include the daily closing price of equity market indices for five emerging markets: Brazil (Bovespa), Russia (MICEX), India (Sensex), China (Shanghai Composite), and South Africa (FTSE/JALSH). The data cover the period from 1 January 2004 to 31 December 2018, making for a total of 3938 observations. These data were retrieved from the Bloomberg database. We computed the daily return for each market as the logarithmic difference in its corresponding market index. The sample of the analysis consists of daily close-to-close returns, which were calculated as the percentage change of the corresponding index:

$$r_t = 100 * \ln\left[\frac{P_t}{(P_t - 1)}\right] \tag{1}$$

where $P_t$ is the closing stock price index for the stock market at time $t$ and $r_t$ is the stock market return. In order to compensate for missing data for a particular market, the time series under study were smoothened out by excluding the corresponding observations in all of other markets, whereas to capture more information on stock prices the daily frequency of data was utilized (Li and Giles 2015). In addition, only common trading days were considered, excluding holidays, weekends, and other non-trading days from the sample.

Table 1 reports descriptive statistics for the stock index of BRICS countries. From these summary statistics, several traits can be identified. Firstly, India has the highest standard deviation among the five BRICS stock markets, followed by South Africa and Russia. Because their standard deviations are much higher than the mean, these stock markets are characterized by higher degrees of volatility. According to the skewness normality test, return distributions for all-time series are negatively and considerably skewed. Furthermore, a high excess kurtosis value implies that all stock return distributions are leptokurtic compared to the normal distribution. The Jarque–Berra test statistics confirm this conclusion, rejecting the hypothesis of normality of stock index returns at the 1% significance level.

In line with descriptive statistics, Figure 1 depicts the visual properties of the daily closing price series for the existence or otherwise of trends, structural breaks, and drifts. The time series plots show the absence of seasonality in all the indices. However, there is evidence suggesting structural breaks in the series given their upward and downward movements. The lower panel shows that all BRICS stock markets exhibit related trends during the period of investigation. Interestingly, until mid-2008, all the stock markets increased continuously. Due to the global financial crisis in late 2008 and 2009, stock markets experienced a sharp fall. Afterwards, these markets experienced an upward trend followed by a downward phase.

Figure 1 shows evidence that the data are not mean reverting. Hence, the data are not stationary. Figure 2 presents the daily returns for the five indices. The attribute of volatility clustering is shown by daily return, as it oscillates around zero. In 2007–2009, higher volatility was demonstrated by all returns.

**Table 1.** Statistical properties of daily return basic descriptive statistics.

|  |  | BOVESPA | MICEX | SENSEX | SHCOMP | JALSH |
|---|---|---|---|---|---|---|
| Level data | Mean | 10.83 | 7.23 | 9.74 | 7.81 | 10.36 |
|  | Median | 10.91 | 7.31 | 9.80 | 7.87 | 10.37 |
|  | Standard Deviation | 0.344 | 0.39 | 0.51 | 0.36 | 0.48 |
|  | Maximum | 11.41 | 7.82 | 10.56 | 8.71 | 11.03 |
|  | Minimum | 9.81 | 6.15 | 8.41 | 6.92 | 9.18 |
|  | Skewness | −1.02 | 1.13 | −0.69 | −0.41 | 0.65 |
|  | Kurtosis | 3.43 | 3.44 | 2.81 | 2.93 | 2.58 |
|  | J–B | 677.63 | 821.85 | 307.49 | 106.26 | 295.13 |
|  | *p*-value | 0.00 | 0.00 | 0.00 | 0.00 | 0.00 |
|  | Observations | 3738 | 3738 | 3738 | 3738 | 3738 |
| Differenced data | Mean | 0.034 | 0.04 | 0.05 | 0.01 | 0.04 |
|  | Median | 0 | 0.07 | 0.02 | 0 | 0.042 |
|  | Standard Deviation | 1.72 | 1.98 | 1.42 | 1.62 | 1.18 |
|  | Maximum | 13.68 | 25.22 | 15.98 | 9.03 | 6.83 |
|  | Minimum | −12.09 | 20.65 | −11.81 | −10.83 | 7.58 |
|  | Skewness | −0.011 | 0.21 | −0.02 | −0.47 | 0.189 |
|  | Kurtosis | 8.92 | 24.19 | 14.02 | 8.27 | 6.95 |
|  | J–B | 5461.08 | 69,966.74 | 18,893.07 | 4463.46 | 2456.56 |
|  | *p*-value | 0.00 | 0.00 | 0.00 | 0.00 | 0.00 |
|  | Observations | 3737 | 3737 | 3737 | 3737 | 3737 |

Note: J–B statistic is the Jarque–Berra normality test. Source: Authors' own calculation.

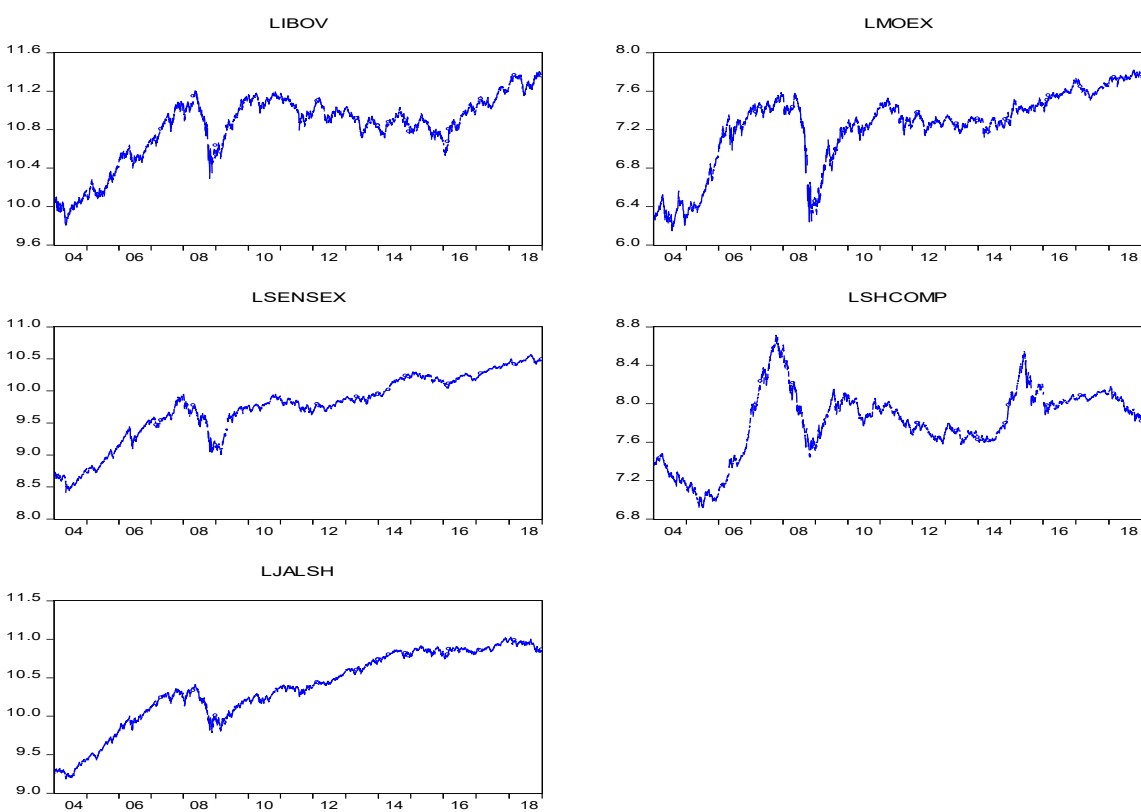

**Figure 1.** Daily closing prices of BRICS stock indices. Source: Authors' own calculation.

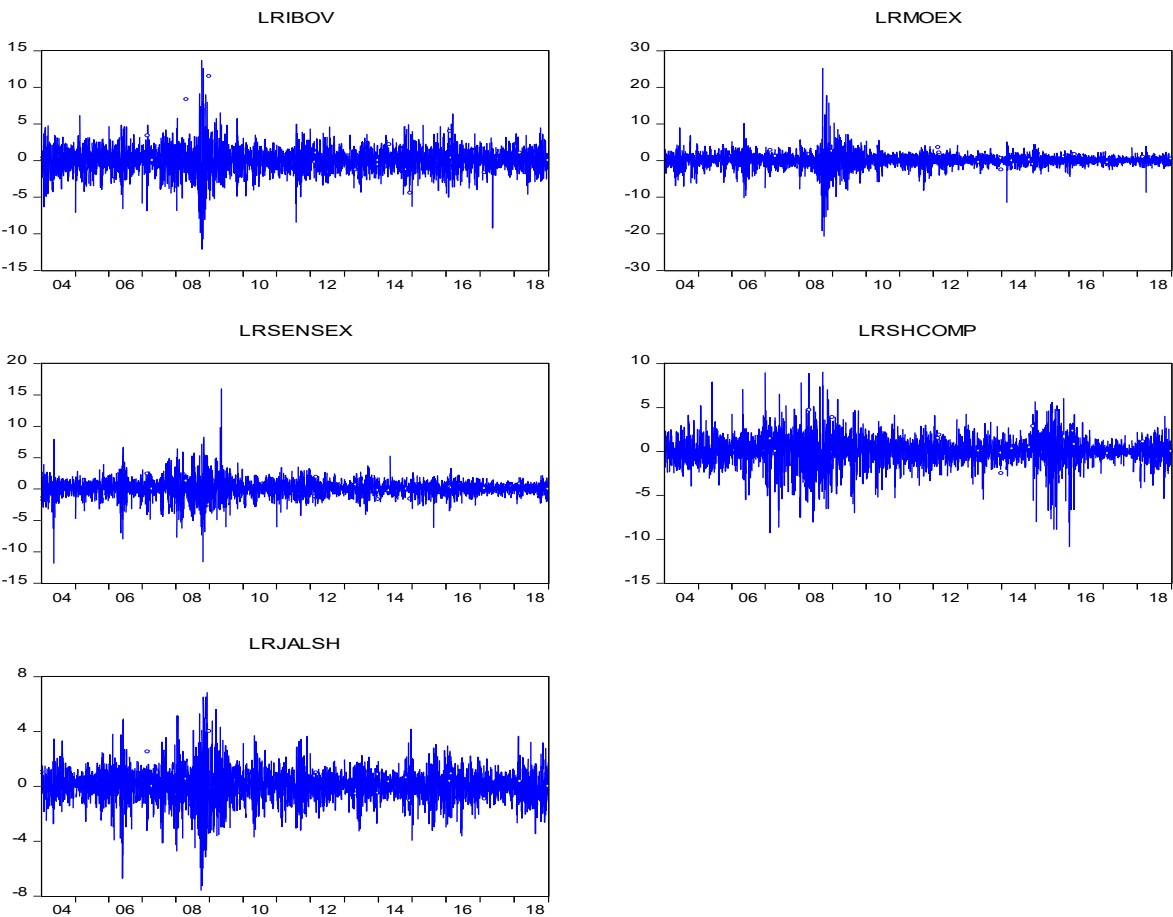

**Figure 2.** Daily return of the BRICS stock indices. Source: Authors' own calculation.

To test for the presence of unit root in the data, we conducted the unit root test with and without a structural break to investigate the integration properties of the series used in the analysis, since we planned to model the various types of cointegration test, which requires that the time series be integrated in the order of one, i.e., I (1). The augmented Dickey and Fuller (ADF; Dickey and Fuller 1979, 1981), Kwiatkowski–Philips–Schmidt–Shin (Shin and Schmidt 1992), and Phillips and Perron (1988) unit root tests were used, which do not consider a structural break, to examine the stationarity of the series and determine the integration order of the non-stationary time series. To examine the stationarity of the series in the event of a structural break, Zivot and Andrews (2002) unit root test was used, which allows for a single structural break. Conventional unit root tests, such as ADF, KPSS, and PP, lose their power and lead researchers to the unreliable conclusion of non-stationarity. The results of the ZA test are consistent with the results of the conventional unit root test.

### 3.2. Nonlinear Threshold Cointegration Test

The main criticism faced by the conventional cointegration methodologies, the Johansen cointegration test (Johansen and Juselius 1990; Johansen 1988) and the ARDL bound test (Pesaran et al. 2001), is that they assume that the cointegrating relationship between the variables does not change over time. This assumption is unrealistic when dealing with long time-series data. Johansen assumes that the series is integrated in the same order while the ARDL test does not take into consideration whether the series is integrated in the same order. Since the series in the study is integrated in the same order, i.e., I (1), the Johansen cointegration test was used to check for the presence of cointegration in the data.

The assumption that the cointegrating vector remains the same between the variables during the entire study period is negated by the presence of a structural break in the time series. Structural breaks in the time series can be due to economic or financial crises,

technological shocks, or policy changes that alter the relationships between the variables (Ghosh and Kanjilal 2014). Several unit root tests have been developed to incorporate the structural break in the unit root test (e.g., Perron 1989; Zivot and Andrews 2002; Bai and Perron 2003), which endogenously determine the break date. The present study uses Zivot and Andrews' unit root test to determine the break date endogenously. For the sake of brevity, the Johansen cointegration methodology is not explained in the present article.

Gregory and Hansen (1996a, 1996b) and Hatemi-J (2008), G–H and H-J henceforth, have argued that structural breaks in a long time series are common phenomena whose presence can change the cointegrating relationship. In other words, the long-run relationship is likely to experience one or two regime shifts in the sample period. In this case, the conventional cointegration tests stated above may produce misleading or inconclusive results. Our study first uses the Johansen cointegration methodology followed by the G–H and H-J threshold cointegration tests to examine regime-switching behavior in the BRICS stock markets. In this study, we used a bivariate model to examine the long-run relationship between the stock markets, when regime-switching occurs due to economic crises, policy and regime changes, and technological changes. It incorporates a bivariate framework since the Hatemi-J (2008) model gives critical values for up to four independent variables with two regime shifts. Thus, to maintain consistency and robustness in the results Gregory and Hansen's (1996a, 1996b) single structural break model and Johansen's cointegration model have been used in the bivariate framework (Hatemi-J 2008; Wei et al. 2019; Bulut et al. 2022).

An essential prerequisite is to check the order of integration of the variables. To investigate this, the augmented Dickey–Fuller (ADF), KPSS, and Phillip–Perron's (PP) tests are used. Moreover, to test the unit root with a structural break the Zivot–Andrews unit root test was used.

*3.3. Gregory and Hansen Cointegration Test*

In 1996, Gregory and Hansen introduced the cointegration test that allows for the possibility of regime shift due to an endogenous structural break. Gregory and Hansen (1996a, 1996b) considered level shift (C), level shift with trend (C/T), and regime shift (C/S) models to test for cointegration with a structural break. The test proposed by Gregory and Hansen is an extension of Engle and Granger's (1987) procedure that allows for a structural break either in the intercept or in the intercept and the cointegrating coefficient at an unknown time. The residuals-based test proposes for the null hypothesis $H_0$: No cointegration with a structural break against alternative assumptions. The model produced the three following simple specifications with two variables:

Model C—Level Shift:

$$y_t = \mu_0 + \mu_1 \varphi_1 + \alpha x_t + v_t \tag{2}$$

Model C/T—Level shift with trend:

$$y_t = \mu_0 + \mu_1 \varphi_1 + \beta_t + \alpha x_t + v_t \tag{3}$$

Model C/S—Regime Shift:

$$y_t = \mu_0 + \mu_1 \varphi_1 + \alpha_1 x_t + \alpha_2 \varphi_t x_t + v_t \tag{4}$$

Equation (2) is a simple case in which a level shift in the cointegrating relationship is modelled as a change in the intercept, where the slope coefficients are held constant. It states that the cointegration relationship has shifted in a parallel fashion. This parameterization $\mu_0$ represents the intercept before the break or shift and $\mu_1$ the intercept after the shift. In Equation (3), $\beta$ is the coefficient of the trend term t, whereas in Equation (4) $\alpha_1$ denotes the cointegrating slope coefficients before the regime shift and $\alpha_2$ denotes the change in the

cointegrating slope coefficient at the break time. Each model has a dummy variable $\varphi_t$ that allows for a structural break. The dummy variable is defined as:

$$\varphi_t = \begin{cases} 1 \ldots if \ \rightarrow t \succ \tau \\ 0 \ldots otherwise \end{cases} \tag{5}$$

where $\tau$ denotes the first structural breaking point in the series. $\tau$ denotes the relative timing of the structural breakpoint or the regime change point that is previously unknown. The procedure is set to test the null hypothesis of no cointegration in the context of models (2)–(4), where there are dummies for the structural break. The Gregory–Hansen test laid the foundations for residuals-based tests, namely the augmented Dickey–Fuller (ADF), $Z^*_\alpha$, and $Z^*_t$ tests, which are applied to regression errors to test the null hypothesis of no cointegration with a structural break in the variables. The null hypothesis is rejected when the $ADF^*$ is smaller than the corresponding critical value. These statistics are defined as:

$$ADF^* = \inf_{(\tau_1, \tau_2) \epsilon\ T} ADF(\tau_1, \tau_2) \tag{6}$$

$$Z^*_t = \inf_{(\tau_1, \tau_2) \epsilon\ T} Z_t(\tau_1, \tau_2) \tag{7}$$

$$Z^*_\alpha = \inf_{(\tau_1, \tau_2) \epsilon\ T} Z_\alpha(\tau_1, \tau_2) \tag{8}$$

where $T = (0.15n, 0.85n)$. The null hypothesis of no cointegration is tested by running Equations (2)–(4) for each possible structural break for $\tau_1$ and then applying Equations (6)–(8) for regression errors for each possible structural break. The smallest value from Equations (6)–(8) is chosen to compare against the critical value of the single endogenous break point test (i.e., the Gregory–Hansen test) in order to accept or reject the null hypothesis of no cointegration.

### 3.4. Hatemi-J Threshold Cointegration Approach

The conventional model presented relies on the fact that all the parameters of the data generating processes are constant and do not allow for a regime change. The Hatemi-J test extends the Gregory–Hansen test by incorporating a cointegration with a single regime shift structural break. This model is extended to two possible regime shifts, where each break's timing is unknown and is determined endogenously.

Hatemi-J (2008) extended the test to account for two structural breaks on both the intercept and the slope. To take into account the effect of two structural breaks on both the intercept and the slope (two regime shifts), following equation is used:

$$y_t = \alpha_0 + \alpha_1 D_{1t} + \alpha_2 D_{2t} + \beta'_0 x_t + \beta'_1 D_{1t} x_t + \beta'_2 D_{2t} x_t + u_t \tag{9}$$

where $\alpha_0$ is the common intercept, $\alpha_1$ is the intercept over the common intercept with one structural break, $\alpha_2$ is the differential intercept with a second structural break, $\beta'_0$ is the slope coefficient of the independent variable, and $\beta'_1$ and $\beta'_2$ are the differential slope coefficient of the first and second structural breaks, respectively. $D_{1t}$ and $D_{2t}$ are dummy variables defined as:

$$D_{1t} = \begin{cases} 0 \ if \ t \leq [n\tau_1] \\ 1 \ if \ > [n\tau_1] \end{cases}$$

and

$$D_{2t} = \begin{cases} 0 \ if \ t \leq [n\tau_2] \\ 1 \ if \ > [n\tau_2] \end{cases}$$

The unknown parameters $\tau_1 \epsilon (0, 1)$ and $\tau_2 \epsilon (0, 1)$ signify the relative timing of the regime change point, and the contents of the bracket denote the integer. The test statistics are the smallest value of these three tests across all the values for $\tau_1$ and $\tau_2$ with $\tau_1 \epsilon$

$T_1 = (0.15, 0.70)$ and $\tau_2\epsilon = T_1 = (0.15 + \tau_1, 0.85)$. The null hypothesis of no cointegration is tested by running Equation (9) for each possible structural break for $\tau_1$ and $\tau_2$ and then applying Equations (6)–(8) for regression errors for each possible structural break. The smallest value from Equations (6)–(8) is chosen to compare against the critical value of the double breakpoint test developed by Hatemi-J to accept or reject the null hypothesis of no cointegration. The parameters are also estimated by running a regression on Equations (4) and (9) for the Gregory–Hansen and Hatemi-J cointegration test parameters. Since the dependent and independent variables are in logarithmic form, the estimated slope maintains elasticity.

## 4. Empirical Analysis
### 4.1. Unit Root Test

The order of integration of the variables is required at the first stage. Table 2 gives the results of the unit root tests based on the augmented Dickey–Fuller (ADF), Phillips–Perron (PP), and Kwiatkowski–Philips–Schmidt–Shin (KPSS) statistics and the first difference of the variables. In the ADF and PP tests, the null hypothesis of the series has a unit root, and they were thus rejected, while in the KPSS test the null hypothesis of the series remained stationary; it was thus rejected for the level data. The KPSS test was to complement the results of the ADF and PP tests. The result of the test reveals that all the series are I (1) in nature.

**Table 2.** Unit root tests.

|  |  | BOVESPA | MICEX | SENSEX | SHCOMP | JALSH |
|---|---|---|---|---|---|---|
| Level data | ADFc | 0.31 | 0.23 | 0.54 | 0.46 | 0.22 |
|  | ADFτ | 0.39 | 0.28 | 0.32 | 0.84 | 0.41 |
|  | PPc | 0.36 | 0.25 | 0.57 | 0.41 | 0.17 |
|  | PPτ | 0.49 | 0.34 | 0.40 | 0.79 | 0.54 |
|  | KPSSc | 4.03 | 4.28 | 6.74 | 2.26 | 7.12 |
|  | KPSSτ | 1.048 | 0.39 | 0.55 | 0.52 | 0.60 |
| Differenced data | ADFc | 0.0001 * | 0.00 * | 0.0001 * | 0.0001 * | 0.00 * |
|  | PPc | 0.00 * | 0.0001 * | 0.0001 * | 0.0001 * | 0.0001 * |
|  | KPSSc | 0.09 ** | 0.09 ** | 0.08 ** | 0.13 ** | 0.30 ** |

Notes: (*) and (**) indicate significance at the 1% and 5% levels, respectively. ADFc and ADFτ are the standards of the augmented Dickey–Fuller (ADF) test statistics and Phillips–Perron (PP) test statistics when the relevant auxiliary regression contains a constant and a constant and trend, respectively (Kwiatkowski et al. 1992).

The conventional unit root test lacks the power to incorporate a structural break in the series; thus, Zivot–Andrew's test (ZA) was considered to check for the structural break in the series. The result of the ZA test is presented in Table 3 and shows that all the series are stationary after the first difference and that there is a structural break in the series.

**Table 3.** Zivot–Andrews unit root test.

|  | Countries | Break with Trend | Break Date | Break with Intercept | Break Date | Break in Both Trend & Intercept | Break Date |
|---|---|---|---|---|---|---|---|
| Level Data | Brazil | −3.2066 | 24 October 2006 | −3.2052 | 14 March 2012 | −3.35984 | 29 May 2008 |
|  | Russia | —— | —— | −3.2444 | 20 June 2008 | −4.45024 | 22 May 2008 |
|  | India | —— | —— | −3.0186 | 15 June 2006 | −5.05497 | 14 January 2008 |
|  | China | −2.8195 | 10 January 2007 | −3.0488 | 21 April 2006 | −3.58464 | 15 January 2008 |
|  | South Africa | —— | —— | −2.4317 | 28 April 2015 | −4.06016 | 23 May 2008 |
| Difference Data | Brazil | −38.3339 | 3 June 2013 | −38.4119 | 27 January 2016 | −38.4317 | 27 January 2016 |
|  | Russia | −32.3506 | 20 August 2008 | −32.4312 | 24 November 2008 | −32.6221 | 28 October 2008 |
|  | India | −44.1867 | 16 November 2008 | −44.2788 | 1 October 2008 | −44.3127 | 11 May 2006 |
|  | China | —— | —— | −28.8215 | 10 July 2007 | −29.0986 | 17 October 2007 |
|  | South Africa | −27.6269 | 23 July 2008 | −20.7662 | 3 April 2009 | −20.906 | 21 November 2008 |

Notes: The critical values at the 1%, 5%, and 10% significance levels are −5.570, −5.080, and −4.82, respectively. Source: Authors' own calculation.

*4.2. Non-Linear Cointegration Test Results*

Table 4 presents the results of the Johansen cointegration tests. We used the Schwarz–Bayesian (SBC) information criteria to determine the optimal order of lags. The lag length chosen by the SBC was one. Conventional cointegration results showed that none of the pairs of variables among the BRICS stock markets are cointegrated. The null hypothesis of no cointegration among the variables is thus accepted. Hence, without incorporating a structural break in the model the results show the absence of any long-run relationship among the BRICS stock markets and suggest the great benefits of portfolio diversification among these countries. The result of the conventional model motivates us to move to the threshold cointegration test to ascertain a long-run relationship between the variables.

Table 5 depicts the results of the threshold cointegration test with a single break, i.e., the Gregory and Hansen cointegration test, and Table 6 depicts the double break regime shift cointegration test by Hatemi-J (2008) (for detail results see Appendix A Tables A1 and A2). The null hypothesis of no cointegration with a single structural break between Brazil and Russia, Brazil and China, Brazil and South Africa, Russia and Brazil, Russia and China, Russia and South Africa, India and China, China and Brazil, China and India, China and South Africa, South Africa and Brazil, South Africa and Russia, and South Africa and China is accepted for all models, i.e., at constant, at constant and trend, and at regime shift for one breakpoint at the 1%, 5%, and 10% levels of significance for the modified $ADF^*$, $Z_t^*$ and $Z_\alpha^*$ tests. Hence, no long-run relationship is established between these countries, and this means that institutional investors can benefit from portfolio diversification. The null hypothesis of no cointegration is rejected for these pairs of countries Brazil and India, India and Brazil, Russia and India, India and Russia, India and South Africa, and South Africa and India. This indicates the presence of long-run relationships between these countries after a single structural break. Thus, the benefits of diversification cannot be reaped from these countries by institutional investors.

Table 6 presents the results of the H-J threshold cointegration test. We can see the statistics and significance of the $ADF^*$, $Z_\alpha^*$, and $Z_t^*$ tests and reject the null hypothesis of no cointegration between Brazil and India, Russia and Brazil, Russia and India, India and Brazil, India and Russia, India and South Africa, China and Russia, South Africa and Brazil, South Africa and Russia, and South Africa and India at the 1%, 5%, and 10% levels of significance. In addition, there are two structural breakpoints in the cointegration relationships between the countries, which correspond to the ranges of 2007–2009 and 2011–2014, respectively. We conjecture that the first structural breakpoint occurred in 2008, near the subprime mortgage crisis's small burst in September 2008, which was characterized by the fall of Lehman Brothers Holdings Inc. New Century Financial, the second-largest sub-prime mortgage lender in the United States, which ran out of money in early April 2007 and declared that the company could no longer give out loans. This event foreshadowed the onset of the 2008 financial crisis, which hit the BRICS stock markets.

In addition, the estimated test value is much lower than the critical value at the 1%, 5%, and 10% significance levels in absolute terms, and thus the null hypothesis of no cointegration between Brazil and Russia, Brazil and China, Russia and China, Russia and South Africa, India and China, China and Brazil, China and India, China and South Africa, and South Africa and China cannot be strongly rejected, even after two regime shifts in these countries. A long-run relationship is not established in these countries, and portfolio diversification benefits can thus be reaped. These pairs are not integrated with each other due to the heterogeneity of the BRICS countries. The results for China do not show any co-movement with the other markets and are consistent with the results of Mohti et al. (2019).

**Table 4.** Johansen cointegration analysis results with no structural breaks.

| Countries | Null Hypothesis | Trace Statistics | Critical Value | Result |
|---|---|---|---|---|
| Brazil–Russia | r = 0 | 10.04 | 15.49 | No Cointegration |
| | r ≤ 1 | 3.91 | 3.84 | |
| Brazil–India | r = 0 | 7.17 | 15.49 | No Cointegration |
| | r ≤ 1 | 1.98 | 3.84 | |
| Brazil–China | r = 0 | 7.17 | 15.49 | No Cointegration |
| | r ≤ 1 | 2.96 | 3.84 | |
| Brazil–South Africa | r = 0 | 8.71 | 15.49 | No Cointegration |
| | r ≤ 1 | 2.29 | 3.84 | |
| Russia–India | r = 0 | 11.55 | 15.49 | No Cointegration |
| | r ≤ 1 | 2.13 | 3.84 | |
| Russia–China | r = 0 | 10.71 | 15.49 | No Cointegration |
| | r ≤ 1 | 4.15 | 3.84 | |
| Russia–South Africa | r = 0 | 10.54 | 15.49 | No Cointegration |
| | r ≤ 1 | 4.08 | 3.84 | |
| India–China | r = 0 | 6.34 | 15.49 | No Cointegration |
| | r ≤ 1 | 1.61 | 3.84 | |
| India–South Africa | r = 0 | 14.65 | 15.49 | No Cointegration |
| | r ≤ 1 | 4.29 | 3.84 | |
| China–South Africa | r = 0 | 9.86 | 15.49 | No Cointegration |
| | r ≤ 1 | 3.01 | 3.84 | |

Notes: r denotes the number of cointegrating vectors. Critical values are summarized (MacKinnon et al. 1999).
Source: Authors' own calculation.

**Table 5.** Gregory and Hansen (1996a, 1996b) cointegration test.

| Countries | Model | $ADF^*$ | $Z_\alpha^*$ | $Z_t^*$ | Lag | Breaks | Result |
|---|---|---|---|---|---|---|---|
| | C | −3.50 | −24.19 | −2.92 | 1 | 2007 | No Cointegration |
| Brazil–Russia | C/T | −3.84 | −29.66 | −3.19 | 1 | 2007 | No Cointegration |
| | C/S | −3.53 | −25.03 | −3.05 | 1 | 2008 | No Cointegration |
| | C | −4.75 ** | −49.38 ** | −4.36 *** | 2 | 2013 | Cointegration |
| Brazil–India | C/T | −4.81 *** | −50.46 ** | −4.45 | 2 | 2013 | Cointegration |
| | C/S | −4.85 *** | −50.7 ** | −4.54 | 2 | 2013 | Cointegration |
| | C | −2.94 | −17.68 | −2.84 | 0 | 2006 | No Cointegration |
| Brazil–China | C/T | −3.92 | −28.54 | −3.64 | 0 | 2013 | No Cointegration |
| | C/S | −3.15 | −21.59 | −3.08 | 0 | 2006 | No Cointegration |
| | C | −3.13 | −20.29 | −2.65 | 1 | 2012 | No Cointegration |
| Brazil–South Africa | C/T | −3.59 | −25.59 | −3.09 | 1 | 2012 | No Cointegration |
| | C/S | −3.53 | −25.58 | −2.98 | 1 | 2013 | No Cointegration |
| | C | −3.07 | −18.65 | −2.58 | 1 | 2014 | No Cointegration |
| Russia–Brazil | C/T | −4.26 | −36.13 | −3.81 | 1 | 2008 | No Cointegration |
| | C/S | −3.15 | −20.01 | −2.66 | 2 | 2014 | No Cointegration |
| | C | −4.46 ** | −44.14 ** | −4.74 | 2 | 2008 | Cointegration |
| Russia–India | C/T | −4.78 *** | −49.31 ** | −4.77 *** | 2 | 2015 | Cointegration |
| | C/S | −5.20 ** | −53.02 ** | −5.17 ** | 0 | 2008 | Cointegration |
| | C | −3.21 | −17.99 | −3.12 | 0 | 2008 | No Cointegration |
| Russia–China | C/T | −3.72 | −24.96 | −3.68 | 0 | 2008 | No Cointegration |
| | C/S | −3.41 | −19.53 | −3.32 | 0 | 2014 | No Cointegration |
| | C | −3.44 | −23.71 | −3.32 | 0 | 2012 | No Cointegration |
| Russia–South Africa | C/T | −3.78 | −28.17 | −3.53 | 0 | 2016 | No Cointegration |
| | C/S | −3.62 | −26.69 | −3.57 | 0 | 2008 | No Cointegration |
| | C | −4.84 ** | −50.84 * | −4.49 ** | 2 | 2013 | Cointegration |
| India–Brazil | C/T | −4.33 | −41.21 | −3.94 | 2 | 2013 | No Cointegration |
| | C/S | −5.20 ** | −58.72 * | −4.81 *** | 2 | 2013 | Cointegration |
| | C | −4.26 | −42.28 ** | −4.61 ** | 2 | 2008 | Cointegration |
| India–Russia | C/T | −4.7 | −50.05 ** | −4.65 | 2 | 2015 | Cointegration |
| | C/S | −4.39 | −44.47 *** | −4.67 | 2 | 2008 | Cointegration |

**Table 5.** *Cont.*

| Countries | Model | *ADF** | $Z_\alpha^*$ | $Z_t^*$ | Lag | Breaks | Result |
|---|---|---|---|---|---|---|---|
| India–China | C | −2.75 | 15.03 | −2.79 | 0 | 2012 | No Cointegration |
| | C/T | −3.48 | −24.54 | −3.57 | 1 | 2015 | No Cointegration |
| | C/S | −2.77 | −15.21 | −2.81 | 0 | 2012 | No Cointegration |
| India–South Africa | C | −4.50 *** | −51.94 * | −5.01 ** | 2 | 2011 | Cointegration |
| | C/T | −5.16 ** | −63.29 * | −5.68 * | 2 | 2011 | Cointegration |
| | C/S | −4.63 | −55.32 ** | −5.14 ** | 2 | 2011 | Cointegration |
| China–Brazil | C | −2.54 | −12.75 | −2.5 | 0 | 2006 | No Cointegration |
| | C/T | −2.67 | −16.12 | −2.6 | 1 | 2014 | No Cointegration |
| | C/S | −2.95 | −16.42 | −2.86 | 0 | 2008 | No Cointegration |
| China–Russia | C | −2.8 | −15.36 | −2.76 | 0 | 2007 | No Cointegration |
| | C/T | −3.44 | −22.35 | −3.42 | 0 | 2007 | No Cointegration |
| | C/S | −3.07 | −19.13 | −3.02 | 0 | 2006 | No Cointegration |
| China–India | C | −2.55 | −13.15 | −2.57 | 0 | 2006 | No Cointegration |
| | C/T | −3.25 | −20.84 | −3.3 | 0 | 2007 | No Cointegration |
| | C/S | −2.73 | −15.22 | −2.75 | 0 | 2010 | No Cointegration |
| China–South Africa | C | −2.9 | −15.58 | −2.84 | 0 | 2011 | No Cointegration |
| | C/T | −2.97 | −16.01 | −2.9 | 0 | 2011 | No Cointegration |
| | C/S | −2.9 | −15.33 | −2.85 | 0 | 2011 | No Cointegration |
| South Africa–Brazil | C | −3.59 | −24.48 | −3.03 | 1 | 2012 | No Cointegration |
| | C/T | −2.82 | −15.65 | −2.71 | 0 | 2006 | No Cointegration |
| | C/S | −4.42 | −37.03 | −3.86 | 1 | 2013 | No Cointegration |
| South Africa–Russia | C | −3.63 | −24.91 | −3.52 | 0 | 2012 | No Cointegration |
| | C/T | −3.88 | −30.67 | −3.7 | 0 | 2016 | No Cointegration |
| | C/S | −3.84 | −27.62 | −3.73 | 0 | 2012 | No Cointegration |
| South Africa–India | C | −4.82 ** | −55.79 * | −5.30 * | 2 | 2011 | Cointegration |
| | C/T | −5.01 ** | −59.45 * | −5.50 * | 2 | 2011 | Cointegration |
| | C/S | −5.22 ** | −64.47 * | −5.72 * | 2 | 2011 | Cointegration |
| South Africa–China | C | −3.64 | −21.64 | −3.62 | 0 | 2011 | No Cointegration |
| | C/T | −3.4 | −19.97 | −3.33 | 0 | 2016 | No Cointegration |
| | C/S | −3.64 | −21.64 | −3.61 | 0 | 2011 | No Cointegration |

Notes: The Gregory and Hansen (1996a, 1996b) test was performed using "ghansen", a STATA module available in the statistical software components archive. The lag length was selected using the Schwartz–Bayesian criterion out of a maximum lag of 7. The break dates were selected automatically by the software. *, **, and *** denote rejection of the null hypothesis at the 1%, 5%, and 10% levels, respectively. Source: Authors' own calculation.

**Table 6.** Hatemi-J (2008) cointegration test with two regime shifts.

| Countries | Break | *ADF* | $Z_\alpha$ | $Z_\tau$ | Lag | Break | Result |
|---|---|---|---|---|---|---|---|
| Brazil–Russia | First Break | −5.527 | −5.61 | −62.194 | 1 | 2007 | No Cointegration |
| | Second Break | | | | | 2012 | No Cointegration |
| Brazil–Sensex | First Break | −5.612 | −79.97 ** | −6.36 ** | 2 | 2009/2011 | Cointegration |
| | Second Break | | | | | 2013/2014 | Cointegration |
| Brazil–China | First Break | −4.241 | −4.22 | −35.476 | 0 | 2006 | No Cointegration |
| | Second Break | | | | | 2014 | No Cointegration |
| Brazil–South Africa | First Break | −6.097 ** | −77.03 ** | −6.186 ** | 1 | 2009 | Cointegration |
| | Second Break | | | | | 2014 | Cointegration |
| Russia–Brazil | First Break | −4.841 | −52.36 *** | −5.173 | 2 | 2007 | Cointegration |
| | Second Break | | | | | 2011 | Cointegration |
| Russia–India | First Break | −5.632 | −77.47 ** | −6.288 ** | 2 | 2008 | Cointegration |
| | Second Break | | | | | 2011 | Cointegration |
| Russia–China | First Break | −3.77 | −27.41 | −3.745 | 1 | 2006/2008 | No Cointegration |
| | Second Break | | | | | 2008/2012 | No Cointegration |
| Russia–South Africa | First Break | −4.55 | −47.39 | −4.794 | 0 | 2007/2008 | No Cointegration |
| | Second Break | | | | | 2007/2009 | No Cointegration |
| India–Brazil | First Break | −5.751 *** | −92.11 * | −6.846 * | 3 | 2011 | Cointegration |
| | Second Break | | | | | 2012 | Cointegration |

**Table 6.** *Cont.*

| Countries | Break | ADF | $Z_\alpha$ | $Z_\tau$ | Lag | Break | Result |
|---|---|---|---|---|---|---|---|
| India–Russia | First Break | −5.638 *** | −80.85 ** | −6.398 ** | 2 | 2008 | Cointegration |
| | Second Break | | | | | 2011 | Cointegration |
| India–China | First Break | −4.195 | −30.19 | −3.86 | 10 | 2006 | No Cointegration |
| | Second Break | | | | | 2012/2011 | No Cointegration |
| India–South Africa | First Break | −6.716 * | −91.35 * | −6.716 * | 8 | 2009 | Cointegration |
| | Second Break | | | | | 2011 | Cointegration |
| China–Brazil | First Break | −3.921 | −31.64 | −3.968 | 1 | 2006 | No Cointegration |
| | Second Break | | | | | 2006/2007 | No Cointegration |
| China–Russia | First Break | −5.246 | −79.44 ** | −6.316 ** | 6 | 2006 | Cointegration |
| | Second Break | | | | | 2006 | Cointegration |
| China–India | First Break | −4.632 | −46.28 | −4.793 | 9 | 2006 | No Cointegration |
| | Second Break | | | | | 2006 | No Cointegration |
| China–South Africa | First Break | −4.38 | −38.39 | −4.38 | 0 | 2006 | No Cointegration |
| | Second Break | | | | | 2006 | No Cointegration |
| South Africa–Brazil | First Break | −5.949 *** | −72.21 *** | −6.116 ** | 1 | 2009 | Cointegration |
| | Second Break | | | | | 2012 | Cointegration |
| South Africa–Russia | First Break | −5.406 | −64.37 *** | −5.688 *** | 0 | 2008 | Cointegration |
| | Second Break | | | | | 2010 | Cointegration |
| South Africa–India | First Break | −5.519 | −88.59 ** | −6.831 * | 8 | 2011 | Cointegration |
| | Second Break | | | | | 2011/2012 | Cointegration |
| South Africa–China | First Break | −4.674 | −43.36 | −4.669 | 0 | 2006 | No Cointegration |
| | Second Break | | | | | 2010 | No Cointegration |

Notes: The Hatemi-J (2008) test was performed using "CItest2b", a GAUSS module written by Hatemi-J (2009) available in the statistical software components archive. The lag length was selected using the Akaike information criterion out of a maximum lag of 5. The break dates were selected automatically by the software. *, **, and *** denote the rejection of the null hypothesis at the 1%, 5%, and 10% levels, respectively. The asymptotic critical values are from Hatemi-J (2008), Source: Authors' own calculation.

### 4.3. Results of G–H Single Structural Break Model Parameters

We also estimated the parameters by running the regression presented by Equation (4) (parameters of single structural break model) and Equation (9) (parameters of H-J double structural break model). The dependent variable is the log of the Bovespa Index $y_t$, and the independent variable is the log of the Sensex index $x_t$ (for one of the pairs). The estimated slope represents the elasticity since all the variables are in logarithmic form. Table 7 shows the parameter estimates obtained from the G–H single structural break model. The crucial inferences drawn from the G–H estimated parameters are: First, Brazil has a significant impact on the Indian stock markets. The estimated elasticity of the Bovespa index with regard to Sensex is equal to 0.822. Secondly, the speed of adjustment is 0.7% between Russia and India. In addition, the estimated elasticity of the Micex index with regard to Sensex is equal to 0.844. The elasticity decreased during the first break period by 0.095. Thus, the integration between the markets decreased after the first break, whereas for India and Brazil the estimated elasticity equals 1.08. Thirdly, the estimated elasticity of the Sensex index with regard to the Bovespa index is equal to 1.078. In the long run, the break period increased by 4.087, and its interaction with the independent variable decreased by 0.0327. In addition, in the short run, the intercept term is significant, and the elasticity of the Sensex index with regard to Bovespa is 0.2437. This suggests that the co-movement between the markets is improved in the long run.

However, the speed of adjustment factor is negative and significant; it changes the disequilibrium between the two markets with the structural break changed at about 0.61%. The elasticity of the Sensex index with respect to the Micex Index is equal to 0.901, whereas in the long run a structural break has no effect on the stock market of India and South Africa. In the short run, the break period increased to 1.638, which implies benefits for institutional investors in the short run. The elasticity of the Sensex index with respect to the interaction of the Jalsh index after the first break is reduced to 0.157. This suggests that the co-movement of the stock markets of India and South Africa is increased in the long run.

Hence, diversification is an advantage in the short run, which is comparatively reduced in the long run.

**Table 7.** G–H cointegration test results: the estimated values of parameters.

| | Countries | Long-Run | | | | Short-Run | | | |
|---|---|---|---|---|---|---|---|---|---|
| | | ECT | $\alpha_1$ | $\beta_0$ | $\beta_1$ | $\alpha_0$ | $\alpha_1$ | $\beta_0$ | $\beta_1$ |
| Model 1 | Brazil–India | −0.120 * | −1.973 | 0.822 * | 0.155 | 0.036 * | —— | —— | —— |
| | | (−4.53) | (−0.87) | (11.46) | (0.69) | (3.48) | —— | —— | —— |
| Model 2 | Russia–India | −0.008 * | 0.667 | 0.844 * | −0.095 | −0.006 | −1.600 * | 0.343 * | 0.192 * |
| | | (−3.09) | (0.31) | (4.71) | (−0.42) | (−0.44) | (−3.96) | (10.75) | (4.52) |
| Model 3 | India–Brazil | −0.116 * | 4.087 ** | 1.078 * | −0.033 *** | −0.241 * | 1.405 * | 0.244 * | −0.129 * |
| | | (−5.44) | (2.24) | (16.67) | (−1.96) | (−2.62) | (4.41) | (15.91) | (−2.62) |
| Model 4 | India–Russia | −0.006 * | −1.239 | 0.901 * | 0.221 | 0.018 ** | —— | 0.236 * | —— |
| | | (−3.73) | (−0.84) | (6.61) | (1.08) | (2.40) | —— | (21.29) | —— |
| Model 5 | India–South Africa | −0.012 * | −2.543 | 1.177 * | 0.220 | −0.028 * | 1.639 * | 0.521 * | −0.157 * |
| | | (−5.17) | (−1.46) | (17.02) | (1.35) | (−2.85) | (4.01) | (24.37) | (−2.85) |
| Model 6 | South Africa–India | −0.011 * | 2.225 *** | 0.756 * | −0.201 *** | 0.308 * | —— | 0.333 * | —— |
| | | (−4.26) | (1.94) | (13.45) | (−1.75) | (3.87) | —— | (26.66) | —— |

Notes: The values in the parentheses are t-values. *, **, and *** indicate significance at 1%, 5%, and 10%, respectively. Source: Authors' own calculation.

*4.4. Results of Hatemi-J Double Structural Break Model Parameters*

Table 8 presents the estimated parameters for the H-J double structural break model. The parameters are estimated by running the regression presented by Equation (9). The dependent variable is the log of the Bovespa Index $y_t$, and the independent variable is the log of the Sensex index $x_t$ (for one of the pairs). All the variables are in logarithmic form, so the estimates represent the elasticity between them. The crucial inference drawn from the H-J estimated parameters are: First, the estimated elasticity of the Bovespa index with regard to the Sensex index is 0.87. The elasticity decreased by 1.47 during the first break period and increased by 2.00 during the second period. Thus, the integration between the financial markets during the second break increased. This empirical finding implies that the potential portfolio diversification benefits have decreased between the two markets, whereas when the impact of the Indian stock market on the Brazil stock market is investigated, the elasticity of the Sensex index with regard to the Bovespa index is 0.91. The elasticity during the first break increased by 4.51 and decreased by 4.80 during the second break. Thus, integration increased during the second break. Secondly, the elasticity of the Bovespa index with regard to the Jalsh index is 1.03. The elasticity reduced to 1.62 during the first break period, increasing to 3.53 during the second period. However, the elasticity of the Jalsh index with regard to the Bovespa index is 0.795. During the first structural break period, the elasticity increased by 0.05 and it decreased by 0.72 during the second break period.

Thirdly, the elasticity of the Micex index with regard to the Bovespa index is 1.46. The elasticity during the first period increased by 0.48 and decreased by 1.29 during the second structural break. Thus, integration between Russia and Brazil reduced during the second period. The same level of integration persists in the short run also. Hence, the benefits of portfolio diversification increased between the two markets. However, in the case of Russia and India, the elasticity of the Micex index with regard to the Sensex index is 0.95. During the first break period, the elasticity increased by 0.16, while it decreased by 0.02 during the second break period. In comparison, the elasticity of Sensex with regard to the Micex index is 0.91. During the first structural break, the elasticity was reduced by 0.06, while it increased by 0.17 during the second break period.

The elasticity of the Sensex index with regard to the Jalsh index is 1.13. During the first break period, the elasticity increased to 25.4 and reduced to 26.42 during the second structural break. Thus, the integration between India and South Africa decreased during the second period. Hence, the potential benefits of portfolio diversification increased between

the two markets. The elasticity of the Jalsh index with regard to the Sensex index is 0.76. During the first structural break period, it decreased by 2.78, while in the second break period it increased by 2.54. These results are in contrast to the impact of the Indian stock market on the South African stock market. However, the elasticity of the Shcomp index with regard to the Micex index is 1.189. The elasticity increased by 25.4 during the first break period and decreased by 26.42 during the second break period. Thus, the integration between the financial markets during the second break was reduced.

In contrast, the elasticity of the Jalsh index with regard to the Micex index is 0.594. During the first structural break period, the elasticity increased by 0.01 and it decreased by 0.516 during the second break period. This empirical finding implies that the potential portfolio diversification benefits have increased between the two markets.

**Table 8.** Estimation results for nonlinear threshold cointegration test results.

| Pairs of Countries | | Long-Run | | | | | | Short Run | | | | | |
|---|---|---|---|---|---|---|---|---|---|---|---|---|---|
| | | ECT | $\alpha_1$ | $\alpha_2$ | $\beta_0$ | $\beta_1$ | $\beta_2$ | $\alpha_0$ | $\alpha_1$ | $\alpha_2$ | $\beta_0$ | $\beta_1$ | $\beta_2$ |
| M1 | Brazil–India | −0.01 * | 14.21 * | −20.19 * | 0.87 * | −1.47 * | 2.00 * | 0.04 * | —— | —— | 0.31 * | —— | —— |
| | | (−4.54) | (3.26) | (−3.88) | (13.17) | (−3.31) | (3.84) | (3.29) | —— | —— | (16.4) | —— | —— |
| M2 | Brazil–South Africa | −0.01 * | 16.69 * | −38.02 * | 1.03 * | −1.62 * | 3.53 * | 0.01 | —— | —— | 0.56 * | —— | —— |
| | | (−4.69) | (6.57) | (−4.51) | (9.40) | (−6.56) | (4.54) | (0.38) | —— | —— | (25.72) | —— | —— |
| M3 | Russia–Brazil | −0.01 * | −5.96 | 14.51 * | 1.46 * | 0.48 | −1.29 * | −0.09 * | −2.22 * | 3.12 * | 0.36 * | 0.21 * | −0.28 * |
| | | (−5.13) | (1.59) | (3.65) | (7.81) | (1.4) | (−3.58) | (−3.26) | (−4.40) | (7.15) | (9.52) | (4.45) | (−7.19) |
| M4 | Russia–India | −0.01 * | −1.81 | 4.38 | 0.95 * | 0.16 | −0.45 | −0.02 | —— | —— | 0.46 * | —— | —— |
| | | (−4.10) | (−0.69) | (1.55) | (8.27) | (0.58) | (−1.55) | (−1.40) | —— | —— | (21.53) | —— | —— |
| M5 | India–Brazil | −0.005 * | −49.51 | 53.13 | 1.06 * | 4.51 | −4.80 | −0.01 | —— | 1.38 * | 0.25 * | —— | −0.12 * |
| | | (−3.82) | (−1.38) | (1.48) | (7.87) | (1.38) | (−1.47) | (−1.23) | —— | (4.44) | (15.93) | —— | (−4.43) |
| M6 | India–Russia | −0.01 * | 0.64 | −1.06 | 0.91 * | −0.06 | 0.17 | 0.02 ** | —— | —— | 0.24 * | —— | —— |
| | | (−4.06) | (0.41) | (−0.50) | (8.46) | (−0.25) | (0.59) | (2.80) | —— | —— | (21.32) | —— | —— |
| M7 | India–South Africa | −0.014 * | 6.86 | −8.41 *** | 1.13 * | −0.659 | 0.79 *** | −0.03 * | —— | —— | 0.47 * | —— | —— |
| | | (−5.42) | (1.61) | (−1.91) | (15.51) | (−1.58) | (1.85) | (−2.50) | —— | —— | (26.17) | —— | —— |
| M8 | China–Russia | −0.003 ** | −182.2 *** | 189.71 *** | 1.189 | 25.4 *** | −26.42 *** | −0.002 | —— | −0.984 * | —— | —— | 0.143 * |
| | | (−2.53) | (−1.85) | (1.87) | (1.33) | (1.85) | (−1.86) | (−0.12) | —— | (−9.37) | —— | —— | (9.72) |
| M9 | South Africa–Brazil | −0.006 * | −0.575 | 8.4212 | 0.795 * | 0.05 | −0.72 | 0.01 | —— | 1.19 * | 0.30 * | —— | −0.11 * |
| | | (−3.75) | (−0.06) | (0.80) | (5.76) | (0.06) | (−0.75) | (1.12) | (1.12) | (4.83) | (24.3) | —— | (1.12) |
| M10 | South Africa–Russia | −0.003 ** | −0.059 | 4.283 | 0.594 | 0.01 | −0.516 | 0.02 ** | —— | —— | 0.32 * | —— | —— |
| | | (2.42) | (−0.18) | (1.01) | (2.42) | (0.11) | (-0.90) | (2.33) | —— | —— | (38.78) | —— | —— |
| M11 | South Africa–India | −0.01 * | 27.11 | −24.57 | 0.76 | −2.78 | 2.54 | 0.03 * | —— | —— | 0.33 * | —— | —— |
| | | (−4.54) | (1.58) | (−1.43) | (14.48) | (−1.57) | (1.44) | (4.11) | —— | —— | (26.68) | —— | —— |

Notes: The values in the parenthesis are t-values. *, **, and *** indicate significance at 1%, 5%, and 10%, respectively. M stands for model.

## 5. Summary and Conclusions

This study examines the cointegrated relationships between the BRICS stock markets using the cointegration tests of Gregory and Hansen (1996a, 1996b) and Hatemi-J (2008) with a view to finding relationships after a possible regime shift in long-run relationships between the BRICS stock markets during the period 2004–2018. The results that were generated through the Johansen cointegration test confirmed that there is no long-run relationship between the countries. The findings of the threshold cointegration test reveal

the existence of a long-run association between the stock prices of BRICS nations under one and two endogenous breakpoints, in contrast to previous studies, which ignored the occurrence of structural breaks during the study period. This indicates that the cointegrating relationships were altered twice during the study period, with two regime shifts having occurred. This explains why structural breakdowns during the study period might lead to erroneous results if they are not considered in the cointegration testing model. However, the threshold cointegration test indicates the presence of long-run relationships between some pairs of BRICS stock markets. The break date was determined endogenously by the test, which indicated that two separate regimes are present. One of the important goals of the study was to revisit the BRICS stock markets and investigate whether these markets share a common break. Our results suggests that a common structural break was found in all five markets (i.e., in 2008) when the data were modelled on both trends and intercepts. However, when the stock markets were modelled for breaks with trends only, only Brazil and China showed breaks in 2006 and 2007, respectively, whereas when the break was modelled with intercepts only, Brazil had a break in 2012, Russia in 2008, India and China in 2006, and South Africa in 2015. However, when the break was modelled on the difference in data for breaks with trends, Brazil had a break in 2013, whereas Russia, India, and South Africa showed breaks in 2008 (i.e., during the global financial crisis). When a break was modelled with an intercept, the break date for Brazil shifted to 2016, while for Russia and India it remained in 2008, for South Africa it moved to 2009, and for China it moved to 2007. Meanwhile, for a break with intercepts and trends, Russia, South Africa, India, and China showed a break in and around the global financial crisis while Brazil showed a break in 2016.

The second objective of the paper was to provide evidence of whether regime switching behavior is prevalent in these markets. A conventional cointegration test provides evidence of no co-movement among the bivariate relationships between the BRICS stock markets. However, non-linear endogenous single and double structural break cointegration tests show integration between some of the pairs of markets (Brazil and India, Russia and India, India and Brazil, and India and South Africa), whereas after the Hatemi-J (2008) test, Brazil and South Africa, Russia and Brazil, China and Russia, South Africa and Brazil, and South Africa and Russia showed integration. The regime switching behavior had an impact on the long-run relationships between the stock markets during the study period. Some of the markets were integrated, but due to the presence of structural breaks some markets were not integrated. Moreover, integration is also suggestive of inefficiency among the markets and the benefit of international diversification is limited in BRICS stock markets that are integrated. The benefits of diversification can be achieved in markets where evidence of no cointegration is present. Strong information flow is not present in those markets. Hence, the markets follow the efficient market hypothesis.

While investigating the effect of regime switching behavior among the pairs that were found to be integrated, it was found that a statistically significant relationship exists between Brazil and India, Russia and India, India and Brazil, India and Russia, India and South Africa, and South Africa and India under a single break model and between Brazil and India, Brazil and South Africa, Russia and Brazil, Russia and India, India and Brazil, India and Russia, India and South Africa, China and Russia, South Africa and Brazil, South Africa and Russia, and South Africa and India under a double break model. Some of the pairs of stock markets between the BRICS countries do not show co-movement even after the double regime shift break test, thus indicating that there are several factors that contribute to the absence of cointegration among these countries, even after a double structural break. Firstly, there can be more than two structural breaks, and this may explain why no integration is detected between these pairs. Additionally, another reason could be the heterogeneity in the policies of the BRICS countries.

The parameters estimated for the G–H and H-J tests show an increase in elasticity during the first break and reduced elasticity in the second break period in some of the pairs and vice versa. Thus, the integration of the stock markets in some of the countries increased during the second break period. This study highlights that the potential benefits of diversification have decreased between BRICS markets. The elasticity increased during the second period in the case of Brazil–India and Brazil–South Africa and decreased during the second period in the case of Russia–Brazil, South Africa–India, and China–Russia with the latter situation benefitting portfolio diversification in the international market.

## 6. Implications of the Study

This empirical study provides useful insights into short-run and long-run investment opportunities, whereas from a long-term investment viewpoint, the existence of cointegration among the stock prices of BRICS countries implies that diversifying one's portfolio by simultaneously holding assets in these markets does not significantly curtail unsystematic risk or upsurge long-term rewards. The results of the study suggest that co-movement between these markets becomes more robust during crisis periods than it is during tranquil periods. Furthermore, we found that economic downturns influences stock prices, subsequently promoting international investors to retract their capital. Individual and institutional investors that diversify their portfolios through international equity investment should keep a close eye on the movement of international stock markets in order to formulate more accurate predictions of the integration of stock markets. Regulators should attempt to understand the reasons for long memory in the markets to help improve efficiency. This study shows that the long-run relationship of markets underwent structural changes in all pairs of return series after accounting for breaks. These findings emphasize that omitting the prospects of a structural break can lead to inaccurate conjecture about the extent of integration and can produce misleading results. Our finding suggest that a structural break has deferred an upsurge in all five studies countries. The study's findings provide useful information on short-run and long-run investment opportunities since structural breaks change the dynamics of integration between markets, hence increasing the opportunity for portfolio diversification in some pairs of countries. Since there is no integration between some of the BRICS stock markets, in the short run an investor can spread out his or her portfolio by holding assets in these unintegrated stock markets.

The empirical findings of this study suggest that the potential rewards of portfolio diversification are elevated between some pairs of markets during break periods. Hence, when there is a break in the market and the integration between two markets is reduced, institutional investors can benefit from investing in these markets and to aid their portfolio diversification. These findings have relevance for policymakers and large investors. Thus, the BRICS stock markets are substantially interconnected and are highly associated with extreme disruptions; policymakers should devise strategies to mitigate systematic risks in the economy when one financial market encounters extreme conditions.

This study has highlighted that investors need to utilize risk management measures to mitigate risks stemming from the instability of stock prices caused by extreme shocks. For individual and institutional investors that diversify their portfolio through international equity investment, keeping a close eye on the changes in stock prices in international stock markets is paramount for formulating a more accurate risk–return prediction of stock market performance.

Structural breaks will likely hamper stock markets' ability to disseminate signals to market participants. This disruption will impact BRICS markets' resource allocation and economic growth. As a result, policies and regulations should be adjusted to ensure resilience in the event of a shock or a structural failure. Policymakers should thus devise a cluster of pragmatic and progressive development plans to diversify and sustain the economy. To undertake economic diversification, the relevant authorities should strengthen the role of the private sector, promote entrepreneurial development, improve the investment climate, and cultivate integration into the global economy to help stock markets to avoid

shocks. Investors must be aware of the likelihood that economic and financial data contain significant structural breaks and regime shifts in order to deal with shocks effectively. Neglecting the potential for these breaks when modelling and estimating asset return volatility with conventional econometric models could have grave ramifications for many financial decisions, including asset pricing, risk management, and portfolio selection.

Finally, it is worth mentioning that the stock market evaluation method conferred in this paper is unique and novel but is not exhaustive and that there is still room for more stock market studies. The current study only incorporates single and double structural break models. Further studies may use multiple structural break models such as that of Maki (2012). Further research, for example, may wish to investigate the occurrence of common structural breaks in stock prices and macroeconomic variables in order to better understand the study's findings. In the era of sustainable development, as we move into the decade of Sustainable Development Goals, it is imperative for financial institutions to play their part. Further studies can be taken up in this area that aim to align policies with Sustainable Development Goals, as the finances of companies are considered to be one of the essential components of sustainable development.

**Author Contributions:** A.S. conceptualized the study, developed the methodology, collected the data, analyzed the data, and wrote and edited the manuscript; M.S. and M.A. conceptualized the study, supervised the work, and edited the manuscript; and M.A.S.A.-F. helped with the data analysis and edited the manuscript. All authors have read and agreed to the published version of the manuscript.

**Funding:** This research received no external funding.

**Institutional Review Board Statement:** Not applicable.

**Informed Consent Statement:** Not applicable.

**Data Availability Statement:** The data will be made available upon request from the corresponding author.

**Acknowledgments:** The authors are thankful to the colleagues who helped during the course of conducting this study.

**Conflicts of Interest:** The authors declare no conflict of interest.

## Appendix A

**Table A1.** Gregory and Hansen (1996a, 1996b) cointegration test.

| Countries | Model | $ADF^*$ | $ADF^*$ Break | $Z_\alpha^*$ | $Z_\alpha^*$ Break | $Z_t^*$ | $Z_t^*$ Break | Lag |
|---|---|---|---|---|---|---|---|---|
| Brazil–Russia | C | −3.50 | 21 February 2007 | −24.19 | 13 August 2007 | −2.92 | 13 August 2007 | 1 |
| | C/T | −3.84 | 10 January 2008 | −29.66 | 13 August 2007 | −3.19 | 13 August 2007 | 1 |
| | C/S | −3.53 | 2 June 2008 | −25.03 | 30 May 2008 | −3.05 | 21 October 2008 | 1 |
| Brazil–India | C | −4.75 ** | 20 May 2013 | −49.38 ** | 20 May 2013 | −4.36 *** | 20 May 2013 | 2 |
| | C/T | −4.81 *** | 20 May 2013 | −50.46 ** | 20 May 2013 | −4.45 | 20 May 2013 | 2 |
| | C/S | −4.85 *** | 16 December 2013 | −50.7 ** | 4 February 2014 | −4.54 | 4 February 2014 | 2 |
| Brazil–China | C | −2.94 | 29 December 2006 | −17.68 | 29 December 2006 | −2.84 | 29 December 2006 | 0 |
| | C/T | −3.92 | 22 April 2013 | −28.54 | 19 April 2013 | −3.64 | 19 April 2013 | 0 |
| | C/S | −3.15 | 19 May 2006 | −21.59 | 16 May 2006 | −3.08 | 19 May 2006 | 0 |
| Brazil–South Africa | C | −3.13 | 17 December 2012 | −20.29 | 7 September 2012 | −2.65 | 7 September 2012 | 1 |
| | C/T | −3.59 | 7 December 2012 | −25.59 | 22 February 2013 | −3.09 | 22 February 2013 | 1 |
| | C/S | −3.53 | 20 June 2013 | −25.58 | 20 June 2013 | −2.98 | 20 June 2013 | 1 |
| Russia–Brazil | C | −3.07 | 26 February 2014 | −18.65 | 19 December 2014 | −2.58 | 19 December 2014 | 1 |
| | C/T | −4.26 | 18 September 2008 | −36.13 | 19 September 2008 | −3.81 | 19 September 2008 | 1 |
| | C/S | −3.15 | 22 December 2014 | −20.01 | 12 December 2014 | −2.66 | 12 December 2014 | 2 |
| Russia–India | C | −4.46 ** | 19 September 2008 | −44.14 ** | 19 September 2008 | −4.74 | 19 September 2008 | 2 |
| | C/T | −4.78 *** | 25 November 2015 | −49.31 ** | 26 November 2015 | −4.77 *** | 26 November 2015 | 2 |
| | C/S | −5.20 ** | 22 September 2008 | −53.02 ** | 17 September 2008 | −5.17 ** | 17 September 2008 | 0 |

**Table A1.** *Cont.*

| Countries | Model | $ADF^*$ | $ADF^*$ Break | $Z_\alpha^*$ | $Z_\alpha^*$ Break | $Z_t^*$ | $Z_t^*$ Break | Lag |
|---|---|---|---|---|---|---|---|---|
| Russia–China | C | −3.21 | 1 March 2016 | −17.99 | 2 October 2008 | −3.12 | 2 October 2008 | 0 |
| | C/T | −3.72 | 18 September 2008 | −24.96 | 18 September 2008 | −3.68 | 18 September 2008 | 0 |
| | C/S | −3.41 | 18 November 2014 | −19.53 | 21 November 2014 | −3.32 | 21 November 2014 | 0 |
| Russia–South Africa | C | −3.44 | 18 April 2012 | −23.71 | 7 May 2012 | −3.32 | 7 May 2012 | 0 |
| | C/T | −3.78 | 20 August 2016 | −28.17 | 16 June 2016 | −3.53 | 16 June 2016 | 0 |
| | C/S | −3.62 | 18 September 2008 | −26.69 | 18 September 2008 | −3.57 | 18 September 2008 | 0 |
| India–Brazil | C | −4.84 ** | 20 May 2013 | −50.84 * | 26 May 2013 | −4.49 ** | 26 May 2013 | 2 |
| | C/T | −4.33 | 20 May 2013 | −41.21 | 26 May 2013 | −3.94 | 26 May 2013 | 2 |
| | C/S | −5.20 ** | 5 July 2013 | −58.72 * | 20 June 2013 | −4.81 *** | 20 June 2013 | 2 |
| India–Russia | C | −4.26 | 19 September 2008 | −42.28 ** | 19 September 2008 | −4.61 ** | 19 September 2008 | 2 |
| | C/T | −4.70 | 25 November 2015 | −50.05 ** | 26 November 2015 | −4.65 | 26 November 2015 | 2 |
| | C/S | −4.39 | 12 August 2008 | −44.47 *** | 19 September 2008 | −4.67 | 19 September 2008 | 2 |
| India–China | C | −2.75 | 11 July 2012 | 15.03 | 11 July 2012 | −2.79 | 11 July 2012 | 0 |
| | C/T | −3.48 | 16 May 2006 | −24.54 | 10 March 2015 | −3.57 | 10 March 2015 | 1 |
| | C/S | −2.77 | 26 September 2012 | −15.21 | 11 July 2012 | −2.81 | 11 July 2012 | 0 |
| India–South Africa | C | −4.50 *** | 31 October 2011 | −51.94 * | 28 October 2011 | −5.01 ** | 28 October 2011 | 2 |
| | C/T | −5.16 ** | 1 September 2011 | −63.29 * | 28 October 2011 | −5.68 * | 28 October 2011 | 2 |
| | C/S | −4.63 | 20 December 2011 | −55.32 ** | 1 December 2011 | −5.14 ** | 1 December 2011 | 2 |
| China–Brazil | C | −2.54 | 20 October 2006 | −12.75 | 6 December 2006 | −2.50 | 6 December 2006 | 0 |
| | C/T | −2.67 | 2 October 2014 | −16.12 | 2 October 2014 | −2.60 | 2 October 2014 | 1 |
| | C/S | −2.95 | 13 May 2008 | −16.42 | 22 April 2008 | −2.86 | 22 April 2008 | 0 |
| China–Russia | C | −2.80 | 7 December 2006 | −15.36 | 10 January 2007 | −2.76 | 10 January 2007 | 0 |
| | C/T | −3.44 | 26 February 2007 | −22.35 | 1 February 2007 | −3.42 | 1 February 2007 | 0 |
| | C/S | −3.07 | 3 May 2006 | −19.13 | 2 May 2006 | −3.02 | 2 May 2006 | 0 |
| China–India | C | −2.55 | 20 October 2006 | −13.15 | 10 November 2006 | −2.57 | 10 November 2006 | 0 |
| | C/T | −3.25 | 11 January 2007 | −20.84 | 10 January 2007 | −3.30 | 10 January 2007 | 0 |
| | C/S | −2.73 | 12 April 2010 | −15.22 | 26 March 2010 | −2.75 | 26 March 2010 | 0 |
| China–South Africa | C | −2.90 | 8 November 2011 | −15.58 | 11 October 2011 | −2.84 | 11 October 2011 | 0 |
| | C/T | −2.97 | 8 November 2011 | −16.01 | 11 October 2011 | −2.90 | 11 October 2011 | 0 |
| | C/S | −2.90 | 8 November 2011 | −15.33 | 24 October 2011 | −2.85 | 28 October 2011 | 0 |
| South Africa–Brazil | C | −3.59 | 7 September 2012 | −24.48 | 7 September 2012 | −3.03 | 7 September 2012 | 1 |
| | C/T | −2.82 | 18 April 2006 | −15.65 | 14 April 2006 | −2.71 | 14 April 2006 | 0 |
| | C/S | −4.42 | 19 June 2013 | −37.03 | 20 June 2013 | −3.86 | 20 June 2013 | 1 |
| South Africa–Russia | C | −3.63 | 18 April 2012 | −24.91 | 3 April 2012 | −3.52 | 3 April 2012 | 0 |
| | C/T | −3.88 | 21 September 2016 | −30.67 | 28 June 2016 | −3.70 | 28 June 2016 | 0 |
| | C/S | −3.84 | 18 May 2012 | −27.62 | 24 May 2012 | −3.73 | 24 May 2012 | 0 |
| South Africa–India | C | −4.82 ** | 31 October 2011 | −55.79 * | 28 October 2011 | −5.30 * | 28 October 2011 | 2 |
| | C/T | −5.01 ** | 26 September 2011 | −59.45 * | 28 October 2011 | −5.50 * | 28 October 2011 | 2 |
| | C/S | −5.22 ** | 18 May 2012 | −64.47 * | 1 December 2011 | −5.72 * | 1 December 2011 | 2 |
| South Africa–China | C | −3.64 | 18 October 2011 | −21.64 | 17 November 2011 | −3.62 | 17 November 2011 | 0 |
| | C/T | −3.40 | 18 October 2011 | −19.97 | 28 June 2016 | −3.33 | 28 June 2016 | 0 |
| | C/S | −3.64 | 6 January 2012 | −21.64 | 17 November 2011 | −3.61 | 17 November 2011 | 0 |

Notes: The Gregory and Hansen (1996a, 1996b) test was performed using "ghansen", a STATA module available in the statistical software components archive. The lag length was selected using the Schwartz–Bayesian criterion out of a maximum lag of 7. The break dates were selected automatically by the software. *, **, and *** denote the rejection of the null hypothesis at the 1%, 5%, and 10% level, respectively.

**Table A2.** Panel A: Hatemi-J (2008) cointegration test with two regime shifts.

| Countries | Break | ADF | ADF Break | $Z_\alpha$ | $Z_\alpha$ Break | $Z_\tau$ | $Z_\tau$ Break | Lag |
|---|---|---|---|---|---|---|---|---|
| Brazil–Russia | First Break | −5.527 | 31 May 2007 | −5.61 | 14 August 2007 | −62.194 | 1 June 2007 | 1 |
|  | Second Break |  | 12 September 2012 |  | 12 September 2012 |  | 12 September 2012 |  |
| Brazil–Sensex | First Break | −5.612 | 16 March 2009 | −79.97 ** | 30 June 2011 | −6.36 ** | 30 June 2011 | 2 |
|  | Second Break |  | 30 October 2013 |  | 9 June2014 |  | 9 June 2014 |  |
| Brazil–China | First Break | −4.241 | 19 May 2006 | −4.22 | 15 May 2006 | −35.476 | 15 May 2006 | 0 |
|  | Second Break |  | 2 July 2014 |  | 14 July 2014 |  | 14 July 2014 |  |
| Brazil–South Africa | First Break | −6.097 ** | 8 July 2009 | −77.03 ** | 8 July 2009 | −6.186 ** | 8 July 2009 | 1 |
|  | Second Break |  | 9 June 2014 |  | 9 June 2014 |  | 9 June 2014 |  |
| Russia–Brazil | First Break | −4.841 | 14 August 2007 | −52.36 *** | 14 August 2007 | −5.173 | 14 August 2007 | 2 |
|  | Second Break |  | 3 May 2011 |  | 14 April 2011 |  | 14 April 2011 |  |
| Russia–India | First Break | −5.632 | 11 August 2008 | −77.47 ** | 22 September 2008 | −6.288 ** | 22 September 2008 | 2 |
|  | Second Break |  | 14 February 2011 |  | 28 January 2011 |  | 28 January 2011 |  |
| Russia–China | First Break | −3.770 | 19 May 2006 | −27.41 | 15 May 2006 | −3.745 | 28 October 2008 | 1 |
|  | Second Break |  | 28 August 2012 |  | 28 August 2012 |  | 28 October 2008 |  |
| Russia–South Africa | First Break | −4.550 | 23 October 2008 | −47.39 | 16 August 2007 | −4.794 | 17 January 2007 | 0 |
|  | Second Break |  | 31 July 2009 |  | 29 July 2009 |  | 17 January 2007 |  |
| India–Brazil | First Break | −5.751 *** | 17 August 2011 | −92.11 * | 4 October 2011 | −6.846 * | 4 October 2011 | 3 |
|  | Second Break |  | 4 April 2012 |  | 3 February 2012 |  | 3 February 2012 |  |
| India–Russia | First Break | −5.638 *** | 12 August 2008 | −80.85 ** | 17 September 2008 | −6.398 ** | 17 September 2008 | 2 |
|  | Second Break |  | 9 November 2011 |  | 16 December 2011 |  | 16 December 2011 |  |
| India–China | First Break | −4.195 | 25 May 2006 | −30.19 | 19 May 2006 | −3.86 | 19 May 2006 | 10 |
|  | Second Break |  | 25 April 2011 |  | 9 April 2012 |  | 9 April 2012 |  |
| India–South Africa | First Break | −6.716 * | 20 May 2009 | −91.35 * | 20 May 2009 | −6.716 * | 20 May 2009 | 8 |
|  | Second Break |  | 18 May 2011 |  | 18 May 2011 |  | 18 May 2011 |  |
| China–Brazil | First Break | −3.921 | 27 April 2006 | −31.64 | 2 May 2006 | −3.968 | 2 May 2006 | 1 |
|  | Second Break |  | 18 July 2007 |  | 23 August 2006 |  | 23 August 2006 |  |
| China–Russia | First Break | −5.246 | 19 April 2006 | −79.44 ** | 2 May 2006 | −6.316 ** | 2 May 2006 | 6 |
|  | Second Break |  | 27 June 2006 |  | 16 June 2006 |  | 16 June 2006 |  |
| China–India | First Break | −4.632 | 19 April 2006 | −46.28 | 2 May 2006 | −4.793 | 2 May 2006 | 9 |
|  | Second Break |  | 8 August 2006 |  | 28 July 2006 |  | 18 August 2006 |  |
| China–South Africa | First Break | −4.380 | 19 April 2006 | −38.39 | 2 May 2006 | −4.38 | 2 May 2006 | 0 |
|  | Second Break |  | 18 August 2006 |  | 18 August 2006 |  | 18 August 2006 |  |
| South Africa–Brazil | First Break | −5.949 *** | 31 August 2009 | −72.21 *** | 3 September 2009 | −6.116 ** | 3 September 2009 | 1 |
|  | Second Break |  | 26 April 2012 |  | 8 February 2012 |  | 8 February 2012 |  |
| South Africa–Russia | First Break | −5.406 | 23 October 2008 | −64.37 *** | 17 September 2008 | −5.688 *** | 17 September 2008 | 0 |
|  | Second Break |  | 20 August 2010 |  | 23 August 2010 |  | 23 August 2010 |  |
| South Africa–India | First Break | −5.519 | 21 August 2011 | −88.59 ** | 21 August 2011 | −6.831 * | 21 August 2011 | 8 |
|  | Second Break |  | 6 December 2011 |  | 20 February 2012 |  | 20 February 2012 |  |
| South Africa–China | First Break | −4.674 | 15 May 2006 | −43.36 | 15 May 2006 | −4.669 | 15 May 2006 | 0 |
|  | Second Break |  | 14 April 2010 |  | 14 April 2010 |  | 14 April 2010 |  |

| Panel B: | Asymptotic | Critical | value |
|---|---|---|---|
|  | 1% CV | 5% CV | 10% CV |
| $ADF^*$ | −6.503 | −6.015 | −5.653 |
| $Z_t^*$ | −6.503 | −6.015 | −5.653 |
| $Z_\alpha^*$ | −90.794 | −76.003 | −52.232 |

Notes: The Hatemi-J (2008) test was performed using "CItest2b", a GAUSS module written by Hatemi-J (2009) available in the statistical software components archive. The lag length was selected using the Akaike information criterion out of a maximum lag of 5. The break dates were selected automatically by the software. *, **, and *** denote the rejection of the null hypothesis at the 1%, 5%, and 10% level, respectively. The asymptotic critical values are from Hatemi-J (2008).

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
