# Peer review of "Are Stock Markets among BRICS Members Integrated? A Regime Shift-Based Co-Integration Analysis"

_economies, doi:10.3390/economies10040087_

Round 1

Reviewer 1 Report

This paper investigates the long-term relationship of stock prices of BRICS countries in a bivariate framework. My major concerns are listed below:

  1. The employed methodology is bivariate. Why is a multivariate model not used? This may serve as a robustness check on the methodology, which is not sufficiently discussed in the current paper.
  2. The empirical data is up to 2018. Is there any reason why more recent analyses, including the COVID recession/recovering periods not investigated?
  3. Section 4.2 is very long and many descriptions are overlapping and can be shortened/summarized. Not all estimation details need to be described.
  4. Similarly, Tables 6 and 7 should be revised to present essential information only. Detailed figures may be presented in the Appendix.
  5. The paper should be copyedited. There are issues of notations and grammatical errors/typos. For instance, in (1), “*” is used instead of “x”.

Author Response

Dear Polina Wang,

Assistant Editor, Economies.

Greetings!

We thank you for considering our manuscript for publication in the journal and for and getting back to us promptly with the required reviews. On the basis of reviewer’s comments, we have carefully addressed all concerns highlighted. The text added and/or modified in the revised manuscript is highlighted in red.

Thank you and we look forward to your response.

Regards,

Authors

Dear reviewers,

Thanks for providing the insightful reviews on our manuscript. The comments highlighted are very important observations and helpful for us to make our manuscript more comprehensive and impactful. We have addressed all the comments and have highlighted them in our revised manuscript.

Thanks once again.

Regards,

Authors  

Reviewer 1

Reviewers Comments

Authors Responses

The employed methodology is bivariate. Why is a multivariate model not used? This may serve as a robustness check on the methodology, which is not sufficiently discussed in the current paper.

Thanks for this important observation; the study applied the Hathemi-J threshold cointegration test, which can be applied, up to four independent variables only (critical value is provided for m=4). So, the analysis of the study was done in bivariate aspect (Since the number of variables involved in the study is five), and that is why we have focussed on the bivariate aspect for the other model (i.e., Johansen cointegration test and Gregory Hansen single structural break model) to maintain the consistency in the analysis (Hathemi-J 2008; Wei et al. 2019; Bulut, Shahbaz and Vo 2021).

In the methodology section, the justification of using bivariate modelling has been added and discussed; please see Section 3.2; please see page no 9. (Highlighted in red)

The empirical data is up to 2018. Is there any reason why more recent analyses, including the COVID recession/recovering periods not investigated?

Thanks for the valuable comment since this study was conceived and conceptualized in 2016-17 as part of PhD research project. The research frame was made keeping in view the data availability and access from the Bloomberg database. Due to limited access to the database, the data was procured for 2004 to 2018 and analysis was carried out accordingly in 2019-20 using various methodological approaches, tools and models.

However, the observation raised by the reviewer is important, and it can be well considered for carrying out any further studies in this field.

Section 4.2 is very long, and many descriptions are overlapping and can be shortened/ summarized. Not all estimation details need to be described.

Thanks for highlighting this, Section 4.2 has been shortened and summarized, please see page no 12. (Highlighted in red).

Similarly, Tables 6 and 7 should be revised to present essential information only.

Detailed figures may be presented in the Appendix.

Thanks for highlighting this, table 6 and Table 7 has been revised with the essential information, however the detailed table with exact date have been provided within appendix. please see page no 13 and 14. (Highlighted in red).

The paper should be copyedited. There are issues of notations and grammatical errors/typos. For instance, in (1), “*” is used instead of “x”.

Thanks for highlighting this important point, the paper has been copyedited.

Reviewer 2 Report

The manuscript deals with a relevant topic in macroeconomics, i.e. the long-term relationship of stock prices in BRICS countries. The manuscript is interesting and well-written, congratulations to the Author(s)!

Below just some minor comments to further raise the overall quality of the paper.

Table 1 can be deleted since it does not add any further information to the reader than those already discussed in the text.

Together with descriptive statistics in Table 2, the Author(s) should report the series of the stock index of BRICS countries in order to allow the reader to carry out a preliminary “visual” inspection of data and to identify possible structural break before proceeding with a more “formal” investigation.

Results section is too much fragmented, above all when the Author(s) present the estimation of parameters of single structural break coefficient for pairs of countries (sections 4.2 and 4.3)

A conclusive section is missing. The Author(s) should present a separate section where policy implications arising from the empirical analysis should be presented and discussed.

Further references for the methodological section:

Barrett, C., Li, J., 2002. Distinguishing between equilibrium and integration in spatial price analysis. Am. J. Agric. Econ. 84 (2), 292–307.

Billio M, Pelizzon L (2003) Contagion and interdependence in stock markets: have they been misdiagnosed? J Econ Bus 55(5):405–426

Koop G, Korobilis D (2016) Model uncertainty in panel vector autoregressive models. Eur Econ Rev 81:115–131

Hamulczuk, M. et al. (2019) “Searching for market integration: Evidence from Ukrainian and European Union rapeseed markets”. Land Use Policy 87: 104078

Author Response

Dear Polina Wang,

Assistant Editor, Economies.

Greetings!

We thank you for considering our manuscript for publication in the journal and for and getting back to us promptly with the required reviews. On the basis of reviewer’s comments, we have carefully addressed all concerns highlighted. The text added and/or modified in the revised manuscript is highlighted in red.

Thank you and we look forward to your response.

Regards,

Authors

Dear reviewers,

Thanks for providing the insightful reviews on our manuscript. The comments highlighted are very important observations and helpful for us to make our manuscript more comprehensive and impactful. We have addressed all the comments and have highlighted them in our revised manuscript.

Thanks once again.

Regards,

Authors  

Reviewer 2:

Reviewers Comments

Authors Responses

Table 1 can be deleted since it does not add any further information to the reader than those already discussed in the text.

Thanks for the comment, this has been addressed by removing Table 1 from the manuscript.

Together with descriptive statistics in Table 2, the Author(s) should report the series of the stock index of BRICS countries in order to allow the reader to carry out a preliminary “visual” inspection of data and to identify possible structural break before proceeding with a more “formal” investigation.

Thanks for highlighting this, the figure has been added giving a visual of raw data of BRICS stock markets and return data has been also added to give a preliminary view of structural break see section 3.1. Please see Figure 1 and Figure 2 on page no 6, 7 and 8 (Highlighted in red)

Results section is too much fragmented, above all when the Author(s) present the estimation of parameters of single structural break coefficient for pairs of countries (sections 4.2 and 4.3)

Thanks for highlighting this, sections 4.2 and 4.3 have been revised, reviewed, and reduced in size. Kindly see page no 12, 16 and 17. (Highlighted in red)

A conclusive section is missing. The Author(s) should present a separate section where policy implications arising from the empirical analysis should be presented and discussed.

Thanks for bringing out this important observation, we have now added two separate sections 5 (conclusion) and 6 (implications). Please see page Nos 21,22, highlighted in red.

Further references for the methodological section:

Barrett, C., Li, J., 2002. Distinguishing between equilibrium and integration in spatial price analysis. Am. J. Agric. Econ. 84 (2), 292–307.

Billio M, Pelizzon L (2003) Contagion and interdependence in stock markets: have they been misdiagnosed? J Econ Bus 55(5):405–426

Koop G, Korobilis D (2016) Model uncertainty in panel vector autoregressive models. Eur Econ Rev 81:115–131

Hamulczuk, M. et al. (2019) “Searching for market integration: Evidence from Ukrainian and European Union rapeseed markets”. Land Use Policy 87: 104078

Thanks for highlighting this, the reference has been incorporated in the manuscript. please see page no. 3 and 4 (Highlighted in red)

Round 2

Reviewer 1 Report

I thank the authors to revise the paper according to my comments. The paper is indeed improved substantially. I recommend publication as in present form.